# Film Forming Systems for Delivery of Active Molecules into and across the Skin

**DOI:** 10.3390/pharmaceutics15020397

**Published:** 2023-01-24

**Authors:** Elka Touitou, Hiba Natsheh, Jana Zailer

**Affiliations:** The Institute for Drug Research, School of Pharmacy, Faculty of Medicine, The Hebrew University of Jerusalem, Ein Karem, Jerusalem 9112102, Israel

**Keywords:** film, derma/transdermal delivery, weblike film, cannabidiol, hemp seed oil, phospholipid, skin, prolonged release

## Abstract

We have investigated delivery systems that can form a structured matrix film on the skin after their application. In a previous work, we have shown that Weblike film forming systems (also called Pouches Drug Delivery Systems, PDDS) enable enhanced skin delivery of the incorporated molecules. These delivery systems are composed of one or more phospholipids, a short-chain alcohol, a polymer and optionally water. In this work, we continue the investigation and characterization of Weblike carriers focusing on some factors affecting the delivery properties such as components concentration and mode of application on the skin. Upon non-occluded application on the skin, the systems dry rapidly, forming a web-like structured film. Lidocaine, Ibuprofen, FITC and Cannabidiol are molecules with various physico-chemical properties that were incorporated in the carrier. The systems were tested in a number of in vitro and in vivo experiments. Results of the in vitro permeation of Ibuprofen through porcine skin indicated two-fold delivery through the skin of Ibuprofen when applied from our Weblike system in comparison with a nanovesicular carrier, the ethosome. We also have investigated weblike systems containing hemp seed oil (HSO). This addition enhanced the film’s ability to deliver lipophilic molecules to the deeper skin layers, leading to an improved pharmacodynamic effect. In analgesic tests carried out in a pain mice model following one hour application of CBD in Weblike system with and without HSO, the number of writhing episodes was decreased from 29 in the untreated animals to 9.5 and 18.5 writhes, respectively. The results of our work open the way towards a further investigation of Weblike film forming systems containing drugs for improved dermal and transdermal treatment of various ailments.

## 1. Introduction

The skin is the largest organ of the body and protects it from exogenous materials. The outermost layer of the skin, the stratum corneum (SC), forms an effective barrier preventing the passage of therapeutic molecules. It is known that the formulation carrier plays an important role in the extent of the active ingredient penetration into the skin layers [1].

In this sense, delivery systems based on nanoparticles, microemulsions, nanoemulsions, film forming systems and phospholipid nanovesicular carriers such as liposomes, ethosomes, transfersome, glycerosome and transethosomes, have been investigated for efficient dermal/transdermal delivery of active agents [2,3,4].

Film forming systems present a promising strategy to improve the delivery properties of topical and transdermal formulations. In general, this type of formulation comprises the active molecule, a film forming polymer and a volatile solvent [5]. Upon contact with the skin, the solvent evaporates leaving a polymeric film containing the active agent. The film obtained increases the contact time between the drug and the skin, allowing a prolonged release of the active agent [6,7] resulting in a less frequent dosing need [8]. Cellulose derivatives, chitosan, polyvinyl pyrrolidine, polyvinyl alcohol polymethacrylate copolymers, and polyacrylate copolymers are often used as film forming agents [9]. Ethanol, isopropanol and butanol serve as solvents in these compositions. Plasticizers, such as glycerin, polyethylene glycol and propylene glycol, can be added to improve the produced films [10].

One advantage of the formed film is its resistance to wiping off and mechanical removal. This is in contrast to other topical dosage forms, such as creams or ointments, which are easily removed from the skin. Furthermore, in general, film forming systems do not leave a sticky and greasy feel after application, leading to improved patient compliance.

Dermal/transdermal film forming systems containing various active molecules such as Betamethasone, Terbinafine hydrochloride, Ropivacaine, Miconazole nitrate, Ketorolac, Methylphenidate, and Vitamin E have been tested in a number of in vitro and in vivo studies for pain management, local anesthesia, and treatment of fungal skin infections, skin inflammation and burn wounds [7,11,12,13,14,15,16]. Saudagar [7] prepared film forming systems of Terbinafine hydrochloride containing ethanolic solution of Eudragit RS PO and hydroxypropylcellulose. The optimized system showed an in vitro antifungal activity of 99.44%. Later, Saudagar and Gangurde [11] incorporated Miconazole nitrate into the same film-forming carrier and obtained a fungal activity of 98.78%. In another work, Ammar et al. [12] used Eudragits^®^ (RLPO, RSPO and E100), polyvinyl pyrrolidone K30 and various penetration enhancers to obtain film-forming solutions for Transdermal delivery of Ketorolac for pain management. The formulations containing the penetration enhancers Lauroglycol^®^ 90 or Azone^®^ showed an improved analgesic effect of the drug by up to 1.5-fold compared to an orally administrated drug in the hot plate test in rats.

In a previous work, we investigated delivery systems that can form a structured film on the skin after their application. In an initial publication on these systems, we introduced the Pouch Delivery System (also called Weblike carrier) for enhanced dermal/transdermal drug delivery. Phospholipids, a short chain alcohol, a polymer, and optionally water are the main components of this carrier. When applied to the skin, these formulations, containing phospholipid reverse micelles, dry, rapidly forming associates of phospholipid reservoirs organized in a web-like (or Pouches) structured matrix film. The work proved the concept behind using reverse phospholipid micelles in a matrix film-forming delivery carrier [17].

The idea behind this work was to study Weblike film-forming formulations differing in their component concentrations and containing various active molecules. For this purpose, several Weblike film forming formulations were prepared and tested in a number of in vitro and in vivo experiments. The in-vitro release profile of Lidocaine from the Weblike systems was tested through a synthetic membrane. Profiles of Lidocaine and Ibuprofen delivery into and across the skin from different Weblike systems were obtained using Franz diffusion cells, porcine ear skin and HPLC quantification. The effect of occlusion was studied by measuring the skin delivery profile of FITC applied in a Weblike carrier under various occlusive conditions. We also present Weblike systems containing hemp seed oil (HSO), which we call Cannaweb systems. Further, the effect on skin penetration of CBD from Cannaweb and Weblike systems was evaluated by measuring its antinociceptive effect.

## 2. Materials and Methods

### 2.1. Materials

The following materials were used: Hydroxypropylecellulose (Klucel ^®^ HF, Hercules, CA, USA), Phospholipon 90G (PL90G, Lipoid, Ludwigshafen am Rhein, Germany), Hemp Seed Oil (HSO, Pukka, Keynsham, UK), Ethanol Absolute or Ethanol 96 (ETOH, Gadot, Netanya, Israel), Propylene glycol (PG, Tamar, Rishon Lezion, Israel), dl-a- tocopheryl acetate (Vit E, Tamar, Rishon Lezion, Israel), Cannabidiol (CBD, STI pharmaceuticals, Essex, UK), Ibuprofen (Sigma, Jerusalem, Israel), Lidocaine Base (Trima, Ma′abarot, Israel), Fluorescein-5-isothiocyanate (FITC, Sigma, Jerusalem, Israel), Tween 80 (Sigma, Jerusalem, Israel), Cetyl alcohol (Holland Moran, Yehud-Monosson, Israel): Castor oil (Holland Moran, Yehud-Monosson, Israel), Cannabigerol internal standard (CBG Standard 1000 μg/mL in methanol 1 mL/ampule, Silicol, Or Yehuda, Israel), Carbopol 980 (Lubrizol, Wickliffe, OH, USA), Ammonia Solution 25% *v/v* (NH_4_OH, Merck, Rehovot, Israel).

### 2.2. Animals

All procedures on animals were carried out according to the National Institute of Health’s regulations and were approved by the Committee for Animal Care and Experimental Use of the Hebrew University of Jerusalem (MD-11-12825-4, MD-17-15076-5)

The experiments were performed on CD-1 ICR mice (24–26 g) or on adult male Sprague–Dawley (SD) rats weighing about 200 g (Harlan, Rehovot, Israel). The animals were housed under standard conditions of light and temperature in plastic cages in the specific-pathogen unit (SPF) of the pharmacy school at the Hebrew University with unlimited access to water and food.

### 2.3. Methods

#### 2.3.1. Preparation of Weblike Film Forming Systems

A weblike carrier comprises phospholipid, ethanol, a film-forming polymer and optionally water. Additives such as glycols, Vit E and ethyl acetate can also be part of the Weblike carrier [17,18]. The Weblike carrier that we call Cannaweb contains added HSO to enhance the dermal and transdermal delivery of lipophilic molecules such as cannabinoids [19]. Weblike carrier formulations tested in this work are presented in Table 1. 

For the preparation of Weblike carriers, the phospholipid was dissolved in ethanol in a covered container. When present, water was then added slowly in a fine stream with constant mixing with an overhead stirrer (Heidolph digital 2000 RZR-2000, Germany). Klucel ^®^ HF was dispersed in the solution while mixing at the same speed. The system was left to repose and then remixed [17,18]. For Cannaweb carrier preparation, phospholipid was dissolved in ethanol, then HSO was added. The film-forming agent, Klucel ^®^ HF was added, through mixing [19].

Weblike and Cannaweb film-forming delivery systems containing molecules with various properties were tested (Table 2). In this work, we present data on systems containing Lidocaine base (log P 2.3), Ibuprofen (log P 3.5), CBD (log P 6.5) or FITC (log P4.8).

To prepare these systems, the active molecules or the fluorescent probe are dissolved in the ethanolic solution of phospholipid. Then the preparation is continued as detailed above.

#### 2.3.2. Characterization of the Weblike System

##### Microscopic Examination of Weblike Structure

The web-like structure of the film generated by the carriers (F8, F9) and systems containing active molecules (F18, F19) was examined by light microscopy, scanning electron microscopy (SEM) and cryo-scanning electron microscopy (cryoSEM).

For light microscopy, samples of 100 µL were spread on a microscopic slide (76 × 26 × 1 mm, Marienfeld, Lauda-Konigshofen, Germany), exposed to air for 2 min, then visualized by light microscopy (Nikon Eclipse TI, Tokyo, Japan), using ×40 objective lens and captured by T-P2 Nikon camera (Japan).

For SEM examination, 30 µL samples of Weblike carrier (F4), CBD Weblike system (F18) and of a control carrier containing ETOH and Klucel^®^ HF and lacking phospholipid were spread on plastic coverslips, dried and coated with Palladium by a sputter coater (SC7620, Quorom, East Sussex, UK). The samples were then examined by FEI quantal 200 SEM at acceleration voltages of 5–10 kV.

Weblike carrier (F2) imaging by cryo-SEM was carried out using specimens prepared with a BAF-060 system (BalTec AG, Liechtenstein). A small drop of the sample was placed on an electron microscopy copper grid and sandwiched between two gold planchettes. The “sandwich” was plunged into liquid ethane at its freezing point, transferred into liquid nitrogen, and inserted into a sample fracture block, pre-cooled by liquid nitrogen. The block was split-open to fracture the frozen sample drop. A Pt-C conductive thin film of 4 nm was deposited on the surfaces (at a 90° angle). The coated specimens were transferred under vacuum by a BalTec VCT100 shuttle, pre-cooled with liquid nitrogen, into a Zeiss Ultra Plus HR-SEM, maintained at −150 °C [20].

##### Size Distribution Test of Phospholipid Reverse Micelles

In this experiment, the mean size of the reversed micelles in solutions of 5% *w/w* PL90G in 60, 65 or 70% ethanol was measured. The mean size distribution was measured in triplicates in an automatic mode by Dynamic Light Scattering (DLS) using a computerized Malvern Zetasizer-nano, ZEN 3600 (Malvern Instruments, Malvern, UK). The determination was carried out at an angle of 173° at 25 °C [21].

#### 2.3.3. In Vitro Release Profile of Lidocaine from the Weblike System

The release profile of a lipophilic molecule, Lidocaine, from the Weblike system formulation (F11) through a synthetic membrane (Phenex nylon membrane 0.20 μm, Phenomenex, Torrance, CA, USA) was measured in-vitro in Franz diffusion cells with an effective diffusion area of 0.64 cm^2^ and 5 mL receiver compartment (PermeGear V6-CB V-Series, PremeGear, Hellertown, PA, USA).

Twenty-five microliters of the formulation were applied non-occlusively to the membrane. The receiver medium, 1:1 *v*/*v* ethanol: water solution, was maintained at 37 ± 0.5 °C and stirred continuously. Receiver samples of 270 µL were withdrawn after 10, 30, 60 and 90 min and replaced with an equal volume of fresh receiver fluid [22].

The concentration of Lidocaine in the samples was quantified by reverse phase HPLC assay (as described in Section 2.3.9).

#### 2.3.4. Skin Penetration of Drugs from Weblike Delivery Systems: In Vitro Tests

The delivery of Lidocaine and Ibuprofen into and across the skin was tested following non-occlusive application to the skin using Franz diffusion assembly and HPLC analysis. Full-thickness clipped porcine ear skin (Lahav, Israel) was used in all the in vitro skin permeation/penetration studies. The integrity of the skin samples was assessed with a magnifying lens.

The Lidocaine amount delivered to skin from Weblike systems containing various phospholipid concentrations (F10–F12) was measured 30 min after application.

Twenty-five microliters of each system were applied on the stratum corneum side of the skin. The experimental conditions related to Franz diffusion cells were similar to those used in Section 2.3.3.

At the end of the 30 min experiment, the skin was removed and cleaned with 600 µL water and kimwipes (Kimberly-Clark, Mississauga, ON, Canada), the receiver was collected, centrifuged for 10 min at 4000 rpm in a Hermle Z 160 M centrifuge (Hermle Labortechnik, Herteller Spintron Inc., Wehingen, Germany) at 25 °C and filtered through Acrodisc^®^ GHP 0.45µm filters (Pall corporation, Port Washington, NY, USA). Then, Lidocaine was extracted from the skin by immersing the tissue in a 7:3 *v*/*v* ethanol: water solution for 48 h at room temperature. The extraction was carried out in a Gyrotory^®^ water bath shaker (New Brunswick Scientific, Edison, NJ, USA) set at a speed of 5.5. At the end of the extraction period, the skin was ultrasonicated for 10 min at 35 °C using an ultrasonic cleaner bath (MRC, Shanghai, China) and mixed for 1 min by Vortex Genie^®^-2 (Scientific Industries, Bohemia, NY, USA) and then the skin was removed. The extraction solution was centrifuged twice for 10 min at 4000 rpm [23]. Lidocaine concentration in the receiver and in the skin was quantified by HPLC.

For Ibuprofen Weblike system, formulations F13 and F14 containing 5 and 10% Ibuprofen, respectively, were used. Receiver samples of 270 µL were withdrawn after 1, 2, 4, 6, 8, 10, 22 and 24 h and replaced with an equal volume of fresh blank receiver fluid. Further, formulations F14 and F15 were tested to learn the effect of phospholipid concentration on the permeation behavior over a period of 8 h by taking receiver samples at 1, 2, 4, 6 and 8 h time points. Moreover, we have compared Ibuprofen skin permeation over 24 h from the Weblike system (F14) with an ethosomal system [24], both containing 10% drug. The ethosomal nanovesicular system contained Ibuprofen: ETOH: PL90G: Vit E: PG: DDW: Carbopol 980: NH_4_OH at a weight ratio of 10:30:2.5:0.2: 10:44.6: 1.7: 1. The system was prepared as previously described by Shumilov et al. [25]. Briefly, the drug and the phospholipid were dissolved in ethanol in a well-sealed container. Water was then added gradually under continuous mixing with a Heidolph^®^ overhead stirrer at 700 rpm. In a separate vessel the carbomer gel was prepared by dispersing Carbopol 980 in water followed by adding NH_4_OH solution through mixing. The obtained carbomer gel was added to the ethosomal composition and mixed well. The mean size distribution of the ethosomal vesicles was measured according to the protocol described in Section 2.3.2.

Skin cleaning, receiver collection and Ibuprofen extraction from the skin were carried out as described above in this section. Lidocaine and Ibuprofen were quantified by reverse phase HPLC assay (Section 2.3.9.). Permeation kinetic parameters were calculated using the Transderm program [26]. All the in vitro permeation experiments were performed in two replicates.

#### 2.3.5. Skin Penetration of FITC: CLSM Studies

In these experiments, we studied the effect of occlusion and the effect of adding HSO to the Weblike carrier on the penetration of FITC into porcine skin using Franz diffusion assembly and CLSM.

##### Evaluation of the Effect of Occlusion on the Skin Penetration of FITC

The vitro penetration profile of FITC into porcine skin was evaluated following its topical application from Weblike system containing HSO (F7) and various occlusion conditions. Twenty-five microliters of the system were applied on the stratum corneum side of the skin mounted on the diffusion cells. The applied occlusion conditions were as follows: non-occlusive, in which the permeation area remained opened; immediately occlusive application, in which the permeation area was covered with a plastic disc immediately upon the carrier application, and occlusive application two minutes following the application. The receiver medium was constantly stirred and consisted of a 3:7 ethanol: water solution. At the end of the 1 h experiment, the skins were removed, and their surface was carefully washed and wiped. The treated skin area was then optically scanned at 10-μm increments through a confocal laser-scanning microscope (Zeiss LSM 710 laser scanning microscopy system, Zeiss, Oberkochen, Germany), with an air plan ×10 objective lens. For the excitation of the label, the 488 nm laser line was used. During the microscopic examination, each skin sample was divided into 5 × 5 tiles and micrographic images obtained. The fluorescence intensity (arbitrary units) was assessed using ImageJ software.

##### Evaluation of the Effect of HSO on the Ability of Weblike to Enhance the Delivery of FITC

This experiment focused on the evaluation of the effect of HSO in the Weblike Carrier (Cannaweb) on the skin penetration profile of the lipophilic probe, FITC. For this purpose, the probe was incorporated in weblike formulations not containing and containing HSO (F6 and F7, respectively), which were applied non-occlusively to the skin mounted on Franz diffusion cells. The experiment was carried out according to the above protocol.

#### 2.3.6. Test of the In Vitro Skin Permeation Profile of CBD from Cannaweb System

The goal of this experiment was to evaluate the in vitro skin penetration and permeation of CBD delivered from the Cannaweb system (F16) as compared to a Control Emulsion. The test was performed using Franz diffusion cells and LC-MS/MS analysis.

The CBD emulsion used in this experiment contains CBD: Cetyl alcohol, Tween 80: Span 80; Castor oil: water at a weight ratio of 5: 5: 12: 10: 26: 42. CBD was incorporated into the cooled emulsion base, which was prepared by the emulsification method. The test was performed on ear porcine skin using 6 Franz static diffusion cells, with 50% Hydroethanolic solution as the receiver medium. The experiment was carried out in two replicates. Twenty-five microliters from each composition were applied on the epidermal side of the skin under non-occlusive conditions.

Receiver samples of 270 µL were collected at 1, 2, 4, 6 and 8 h, and replaced with an equal volume of fresh blank receiver fluid. After 8 h, the skin surface was carefully washed with 600 µL water and kimwipes (Kimberly-Clark, Canada). The collected receiver samples were centrifuged for 10 min at 4000 rpm. Then, 198 µL receiver supernatant was mixed with 2 µL of the CBG internal standard 10 µL/mL. Then the samples were injected into LC-MS/MS to assess the permeation of CBD across the skin.

To quantify the CBD penetrated through the skin, the tissue was immersed in 5 mL ethanol for 18 h at room temperature. The extraction was carried out in a Gyrotory^®^ water bath shaker (New Brunswick Scientific, Edison, NJ, USA) set at a speed of 6. The extraction solution was diluted at a ratio of 1:10 with a co-solvent composed of acetonitrile: water (80:20), centrifuged for 10 min at 4000 rpm and filtered through Acrodisc^®^ GHP 0.45 µm filters (Pall corporation, Port Washington, NY, USA). Prior to injection in LC-MS/MS, 495 µL of the diluted extraction solution was mixed with 5 µL of the CBG internal standard.

The amount of CBD in HSO used in this experiment was quantified by LC-MS/MS using a modified published method [27]. CBD was extracted from HSO as follows: a volume of 100 µL HSO, blank or spiked with a known amount of CBD in the range of 1–100 ng/mL, was mixed well with 5 µL of the CBG internal standard. Then, 400 µL of acetonitrile were added to each sample and mixed by Vortex^®^ for 3 min, then left to rest for five minutes and mixed again for 3 min. Then, the samples were centrifuged at RT 5 min at 14 k rpm. The supernatants were filtered and transferred into pre-labeled vials prior to injection into the LC-MS/MS apparatus.

#### 2.3.7. Evaluation of the Anti-Nociceptive Effect of the CBD Cannaweb System Applied to the Skin of a Mice Model of Pain

The pharmacodynamic effect of the CBD Cannaweb film-forming system, F16, containing 5% *w*/*w* CBD and 3% *w*/*w* HSO applied to the skin of pain mice model [28] was tested compared to a Weblike system lacking HSO (F 17).

Thirteen animals were divided randomly into 3 groups. One day before the experiment, the dorsal skin area of the animal was clipped (Oster, Owosso, MI, USA). On the day of the experiment, the animals were anesthetized shortly with Isoflurane^®^ and each was treated with 100 mg/kg CBD. Fifty milligrams of F16 (Cannaweb containing 3% HSO, *n* = 5) or F17 (Weblike not containing HSO, *n* = 4) were applied topically and non-occlusively on 1 cm^2^ of the shaved skin area. One hour after treatment, the animals were anesthetized again and injected intraperitoneally with acetic acid (0.6% *v*/*v*) at a dose of (10 mL/kg) (*n* = 4). The third group contained four animals that served as untreated controls. Animals in this group were anesthetized with Isoflurane^®^ and injected with acetic acid at the same dose without drug treatment.

The writhing episodes were recorded by counting the number of writhes 5 min after acetic acid administration for a period of 20 min. Writhes were indicated by abdominal constriction and stretching of at least one hind limb.

The analgesic effect of each treatment is expressed by the Maximum Possible Effect (MPE%) of the treatments, which is directly related to the efficiency of the treatment, and is calculated according to the following equation:

MPE% = [Mean of writhing in control group − number of writhing in each mouse in treated group]/[Mean of writhing in control group] ×100

#### 2.3.8. Evaluation of the Stability of Weblike Carrier

The stability of formulation F2 was determined by monitoring changes in the organoleptic properties, in the viscosity and pH values during 30, 60 and 60 days of storage at room temperature (RT), 4 °C and 40 °C.

The viscosity of Weblike carrier was measured at room temperature (25 °C) using a Brookfield DV-III viscometer coupled to a temperature controlling unit. A cylindrical spindle LV4 was used and a rotational speed of 250 rpm was applied.

The pH of the carrier was measured using Eutech instruments pH 510 (pH/mV/°C meter, Malaysia) equipped with Russell Thermo electron corporation electrode (UK).

#### 2.3.9. HPLC and LC-MS/MS Assays

The amounts of Lidocaine and Ibuprofen in the various samples were quantified by reverse-phase HPLC, using a Merck-Hitachi D-7000 apparatus equipped with an L-7400 variable UV detector, L-7300 column oven, L-7200 auto-sampler, L-7100 pump, and an HSM computerized analysis program (Tokyo, Japan).

Lidocaine assay was carried out at 210 nm, using a Zorbax Eclipse XDB C18, 5 μm 4.6 × 150 mm (Agilent, USA) column, at 30 °C, using acetonitrile: phosphate buffer 0.01 M pH 6 (35:65 *v*/*v*) mobile phase at 1.2 mL/min flow rate.

For Ibuprofen quantification, a LiChospher^®^ 100 C 18 5 μm 4 × 250 mm (Merck, KGaA, Darmstadt, Germany) column at 25 °C, with a mobile phase composed from methanol: acetate buffer 0.03 M pH 4 (80:20 *v*/*v*) at 1 mL/min flow rate was used. The molecule was detected at a wavelength of 225 nm.

The retention time was ~8.8 and ~5 min for Lidocaine and Ibuprofen, respectively.

CBD quantity in the receiver samples, the skin and in HSO was analyzed by LC-MS/MS, Sciex (Framingham, MA, USA) Triple Quad™ 5500 mass spectrometer coupled with a Shimadzu (Kyoto, Japan) UHPLC System. CBG was used as an internal standard. The chromatographic separation was performed on a CORTECS^®^ (Waters Corp., Milford, MA, USA) column (C18, 2.7 µm particle size, 50 × 2.1 mm), protected by a VanGuard^®^ (Waters Corp., Milford, MA, USA).

The injection volume was 5 μL, the oven temperature was maintained at 40 °C and the autosampler tray temperature was maintained at 5 °C. CBD and its internal standard, CBG, were detected in positive ion mode using electron spray ionization (ESI) and multiple reaction monitoring (MRM) modes of acquisition. The molecular ions of the compounds [M + H]+ were selected in the first mass analyzer and fragmented in the collision cell followed by the detection of the products of fragmentation in the second analyzer. The precursors were 315.1 *m*/*z* for CBD and 317 for CBG, and the products were 193.0 and 193.1 *m*/*z* for the two molecules, respectively.

#### 2.3.10. Data Analysis and Statistical Evaluation

The results are expressed as mean values ± standard deviation (SD) or as mean ± standard error of mean (SEM).

Statistical analysis of the data was performed by GraphPad InStat program, using unpaired two-tailed t-test or ANOVA with post-test at a significance level set at *p* < 0.05 or less.

## 3. Results

### 3.1. Microscopic Structure of Weblike Films

Examination of various Weblike films was carried out by light microscopy, SEM and cryoSEM. The micrographs are given in Figure 1.

The above micrographs obtained by various microscopic techniques show the presence of the web-like film structure. Light microscopy indicates the presence of a structured film generated from the Weblike carrier (Figure 1A). These structures were absent in the sample generated from a control not containing phospholipid (Figure 1B). SEM and cryo-SE micrographs presented in Figure 1C,D, respectively, confirm the presence of the film.

Further, the films obtained from Cannaweb and Weblike systems containing CBD (F18 and F19, respectively), were also composed of web-like structures as indicated by SE and light microscopy (micrographs not shown).

### 3.2. Size Distribution of Phospholipid Reverse Micelles

In our previous study on Weblike delivery systems, ^31^P-NMR spectra suggested a reverse micellar organization of phospholipid in high ethanolic solutions [17].

In the present work, we measured the phospholipid micelles size distribution by DLS analysis. The results showed one narrow peak of a homogenous population of reverse micelles with normal size distribution.

The average size of the reverse phospholipid micelles in 65% ethanol solution was 2.98 ± 0.12 nm.

It was also interesting to test the effect of increasing ethanol concentrations on the size of the micelles in the solution. In ranges of 60 to 70% ethanol concentration, the average diameter of the micelles decreased twofold from 4 to 2 nm.

### 3.3. Lidocaine Release Profile from Weblike System

In order to characterize the drug release behavior from the Weblike film, the release of the model lipophilic drug Lidocaine was investigated using a synthetic membrane and Franz diffusion cells. Figure 2 shows the cumulative amount of Lidocaine (initial concentration 9.6%) released from the Weblike system and permeated the membrane over a 90 min period.

The results indicated that the lidocaine release profile fits the Higuchi equation [29]: Q = 2C0(Dt/π)^1/2^

The diffusion model is based on Fickian diffusion from a polymer matrix, where Q is the cumulative amount of drug released per unit surface area, C0 is the initial drug loading, D is the diffusion coefficient of the drug in the matrix and t is the time after the commencement of diffusion.

The plot becomes linear after an initial lag time, indicating a diffusion-prolonged release from the Weblike film. A slope of 4.42% released per min^1/2^ and R^2^ of 0.99 were calculated from the plot in Figure 2. Fick’s law of diffusion provides the fundament for the description of solute transport from a polymeric matrix film [30].

### 3.4. Effect of Various Components Concentration in the Weblike System on Dermal/Transdermal Drug Delivery

The skin penetration profile of model molecules from the Weblike system and control system (without phospholipid) was evaluated in our first publication [17]. For example, a superior delivery of Rhodamine B and FITC into the skin, in terms of intensity and depth, was obtained for the Weblike carrier. These findings confirm the important role of the reverse micelles in the penetration enhancing effect of the Weblike (Pouches) drug delivery carrier.

Consequently, it was interesting to estimate the influence of phospholipid concentrations in the Weblike system on skin permeation and penetration of the lipophilic molecule, Lidocaine base. For this purpose, we have prepared Lidocaine Weblike systems with phospholipid concentrations in the range of 2.5 to 10%.

As shown in Figure 3A,B, raising the phospholipid concentration in the Weblike system from 2.5% to 5% increased Lidocaine accumulation in the skin by 1.5-fold, from 93.19 ± 24.85 to 150.43 ± 56.3 μg/cm^2^. The amount of drug permeating across the skin also increased by 2.5-fold from 0.42 ± 0.31 to 1.08 ± 0.58 μg/cm^2^. On the other hand, the presence of 10% phospholipid concentration in the formulation decreased (*p* < 0.05) the drug amount in the skin to 59.79 ± 9.11 μg/cm^2^ and the amount permeating across the skin to 0.08 ± 0.02 μg/cm^2^.

The effect of phospholipid concentration in the system on the dermal and transdermal delivery of Ibuprofen was also tested. Ibuprofen Weblike systems containing 5 or 10% phospholipid were tested to understand the effect of this component concentration. Ibuprofen’s skin delivery into and across the skin was measured in vitro in Franz diffusion cells 8 h after application. Figure 3C shows that raising the phospholipid concentration from 5 to 10% decreased the drug amount delivered into the skin from 625.94 ± 69.57 to 491.87 ± 143.94 μg/cm^2^, respectively. A similar effect of phospholipids was observed in the receiver, where the permeated Ibuprofen amount dropped from 219.8 ± 83.6 to 118.7 ± 61.8 μg/cm^2^ (Figure 3D). It is noteworthy to mention that this effect of phospholipid concentrations on Ibuprofen delivery into and across the skin was non-significant (*p* ˃ 0.05).

Next, we measured the effect of drug concentration on its in vitro skin permeation profile. For this purpose, we tested formulations containing Ibuprofen 5% and 10%. The 24 h skin permeation profiles of Ibuprofen showed a prolonged permeation behavior, with a lag time phase followed by a steady state flux. A drug flux of 6.64 × 10^−2^ mg/h × cm^2^ was calculated from the linear portion of the plot, for the 10% Ibuprofen Weblike system. This value is two times higher than the Ibuprofen flux measured from the 5% Ibuprofen Weblike system (3.28 × 10^−2^ mg/h × cm^2^). The lag time values were shorter, without statistical significance, for the 10% Ibuprofen Weblike system than for the 5% Ibuprofen system, 99 min vs. 114 min, respectively.

The next step in this work was to test the skin permeation behavior of the Weblike carrier in comparison to another phospholipid skin permeation-enhancing carrier, the ethosome. For this purpose, we measured Ibuprofen skin permeation from the Ibuprofen Weblike system vs. Ibuprofen ethosomes with a mean vesicular size of 250 nm, each containing 10% drug. Results presented in Figure 3F show a 4.82 × 10^−2^ mg/h × cm^2^ flux for Ibuprofen delivered by the Weblike system. This value is more than two times higher compared to the 2.19 × 10^−2^ mg/h × cm^2^ Ibuprofen flux measured from the ethosomal system. Furthermore, 1708.1 ± 324.6 μg/cm^2^ of Ibuprofen permeated through the skin during 24 h when delivered by the Weblike system, which was significantly higher (*p* < 0.05) than the permeated amount of the drug delivered by ethosomes (775.2 ± 262 μg/cm^2^).

### 3.5. Effect of Occlusion on Skin Penetration of Molecules Incorporated in Weblike Carrier: A CLSM Study

In this experiment, we tested if occlusion may affect the delivery to skin performance of molecules incorporated in the weblike film-forming carrier. In a CLSM study, the in vitro skin penetration of FITC was tested using Franz diffusion cells and porcine ear skin. Three protocols have been used: (1) a non-occlusive system application; (2) an occlusive application by covering the skin immediately following the system application, and (3) an occlusive application that has been carried out 2 min after system application to the skin. The results of this in vitro experiment are presented in Figure 4.

The various fluorescence intensity values at their maximum point were 15.1, 12.5 and 21.5 A.U. for non-occlusive, occlusive covered immediately and occlusive covered 2 min after application, respectively. These maximum values were measured at skin depths of 40 microns for the opened and the immediately covered application and 50 microns for the non-immediate occlusive one. The calculated area under the curve (AUC) values were 839.6, 614.7 and 983.5 AU. µm for the opened application, immediately occlusive application and the occlusive after two minutes application, respectively.

Overall, the results of this experiment indicate that non-immediate occlusive application of Cannweb can lead to the best penetration profile of the tested probe. It is possible that this behavior resulted from combined mechanisms; during the first two minutes of application before the occlusion, the high ethanol concentration and the small reverse micelles enhance the probe delivery into the skin. Concomitantly the film is generated then is occluded allowing better penetration into the skin.

### 3.6. Effect of HSO on Skin Penetration of Molecules Incorporated in Weblike Carrier: A CLSM Study

In this in vitro study in Franz diffusion cells, we tested the penetration of FITC into ear porcine skin from two systems: the Cannaweb and the Weblike carrier lacking oil. The results of this test are presented in Figure 5.

A maximum fluorescent intensity of 54.5 A.U. was measured at a skin depth of 50 µm for the Cannaweb system, while the maximum fluorescent intensity for the Weblike carrier not containing the oil was 35.5 A.U. at a skin depth of 40 µm.

The fluorescent signal of FITC reached a skin depth of 120 µm for Cannaweb, compared to only 100 µm for the system lacking the oil. The most intense fluorescence was seen at a depth of ~40–60 microns.

The calculated AUC generate values of 3222.2 vs. 1809 (AU. µm), for Cannaweb and the system not containing the oil, respectively.

### 3.7. In Vitro Skin Permeation Profile of CBD from Cannaweb

The goal of this experiment was to evaluate the permeation and penetration of CBD Cannaweb system into and across the skin in comparison with a conventional topical dosage form, the emulsion. The in vitro results presented in Figure 6, indicate that at the initial time points (1–2 h), Cannaweb and the emulsion system led to similar CBD permeation of 9.3 ± 1.0 and 10.4 ± 3.5 ng/cm^2^, respectively. After this point, the delivery of CBD from Cannaweb started to rise, reaching a value of 114.3 ± 39.8 ng/cm^2^ compared to only 15.4 ± 4.1 ng/cm^2^ for the emulsion. By the end of the 8 h, the cumulative permeated CBD amount from Cannaweb vs. the control emulsion was 1745.2 ± 249.4 and 236.1 ± 71.0 ng/cm^2^, respectively.

The CBD amount extracted from the skin at the end of the experiment was 29.9 ± 5.5 µg/cm^2^ from Cannaweb comapred to 2.6 ± 0.5 µg/cm^2^ for the control system. This penetration was improved by more than 11 fold by means of Cannaweb, emphasizing the important role of the carrier to enhance and prolong the delivery of the incorporated molecule. It was interesting to know if the HSO used in thse experiment contained CBD. For this purpose, we extracted CBD from the oil and quantifed it by LC-MS/MS. We found a negligile CBD concentration of 0.000142%.

### 3.8. Effect of HSO on the Anti-Nociceptive Action of CBD Applied to Mice from Weblike Systems

In this experiment, we tested compositions containing Cannabidiol incorporated in Weblike and Cannaweb film-forming carriers for their analgesic effect in a mice pain model.

The analgesic effect of the CBD applied from Cannaweb (Weblike system containing 3% HSO) and compared to a system not containing HSO. The results presented in Figure 7 indicate that CBD application to male CD-1 ICR mice from Cannaweb system led to a significant analgesic effect 1 h following treatment, with 67.2% MPE. This is in comparison to only 36.0% for an equal dose of CBD applied in the Weblike system not containing HSO.

### 3.9. Stability of Weblike Film Forming Carrier

The accelerated and intermediate term stability of the Weblike carrier was tested by monitoring changes in the organoleptic observations and in viscosity and pH values over time.

No changes in the organoleptic appearance of the systems over a period of up to one year were observed. Results of the viscosity and pH measurements at zero time and after 30, 60 and 90 days at room temperature (RT), 4 °C and 40 °C showed no significant variations in tested parameters of the carrier throughout the storage interval at the tested conditions (Table 3).

## 4. Discussion

It is acknowledged that the carrier for topical application of active molecules to the skin plays a key role in the extent of their delivery into and across the skin.

In this work we have investigated structured film-forming formulations we call “Weblike or Pouches systems”. This system comprises phospholipids, a short chain alcohol, a polymer and optionally water. Following application of the system to the skin, ethanol fluidizes the lipids in the stratum corneum, and the reverse phospholipid micelles carry the active molecules into the skin. Concomitantly, the polymeric film containing phospholipid reservoirs is generated allowing a prolonged delivery into and across the skin.

In our first publication on this structured film, we have shown by ^31^P-NMR studies, that at high concentrations of ethanol (˃60%), phospholipid molecules are organized as reversed micelles. We also studied the in vivo skin penetration of Rhodamine B and FITC after 1 h of non-occlusive application to rats in the film forming carrier by CLSM in comparison to control systems lacking the reverse phospholipid micelles [17]. The obtained results confirmed the important role of these associates for enhanced skin delivery via the film-forming carrier.

Reverse micelles are self-assembled nanoaggregates of amphiphiles that form in a non-aqueous solvent. Structurally, the hydrophilic groups of the surfactant represent the interior core of these associates, and the hydrophobic ones face the external (non-aqueous) medium. These systems have been studied for dermal or transdermal delivery of active molecules including Insulin [31], Imatinib mesylate [32], Hydrophilized Melanoma Antigen Peptide [33] and Hyaluronic acid [34]. Incorporating drugs in reverse micelles could lead to sustained drug release.

In the present work, we prepared and investigated Weblike film-forming formulations that differ in their component concentrations and containing various active molecules. We also presented the Cannaweb carrier, a special form of the carrier containing HSO. The in-vitro release profile of Lidocaine from the Weblike systems through a synthetic membrane indicated a prolonged release behavior from a polymeric matrix film. Profiles of Lidocaine and Ibuprofen delivery into and across porcine ear skin from different Weblike systems, measured in Franz diffusion cells by HPLC quantification, indicated a non-significant effect of phospholipid on the delivery behavior. Non-immediate occlusive application was found to be the optimum occlusive condition for applying Weblike delivery systems, as measured by CLSM. Adding HSO to the system improved by ~2 fold the skin penetration of FITC. Skin permeation and penetration of CBD were improved by more than 7- and 11-fold, respectively, as a result from application in the Cannaweb system, as measured in Franz diffusion cells by LC-MS/MS. Further, the antinociceptive effect of CBD was improved by 2-fold when incorporated in the system containing HSO compared to the oil-free system in a mice model of pain.

The structure of the obtained film was characterized. Light and electronic microscopy revealed the formation of a special web-like structured film. It was further interesting to measure the mean size of phospholipid reverse micelles in the Weblike carrier. DLS data indicate that in hydroethanolic solutions containing more than 65% ethanol, the size of the micelles ranged from 2–4 nm. Increasing the concentration of ethanol in the system resulted in a non-significant decrease in the micellar size. This is in contrast to the changes in the size of phospholipid vesicles as a result of modifications of system composition previously reported for ethosomal vesicles, where the increase in ethanol concentration resulted in smaller vesicles [24].

Further, the in vitro Lidocaine release measured by the permeation rate of the drug through a synthetic membrane showed that the released amount of Lidocaine was proportional to the square root of time. The release slope was 4.42% per min^1/2^ with an R^2^ of 0.99. These results indicate a prolonged release of the drug from the Weblike carrier which fits with Higuchi’s equation from a polymeric matrix film.

The next step was to learn about the skin penetration and permeation behavior of two drugs, Lidocaine and Ibuprofen. This was carried out in a number of in vitro permeation studies in Franz diffusion cell assembly using full-thickness porcine ear skin. The results show that 5% phospholipid concentration in the system led to improved delivery into and across the skin of the two drugs by up to 7-fold. These data suggest that the Weblike system containing 5% phospholipid can be considered a formulation with a good potential for efficiency.

As a next step, it was interesting to examine the delivery properties of the Weblike system in comparison with the nanovesicular carrier, the ethosome. We found a 2-fold increase in the permeation of Ibuprofen in Weblike relative to the ethosomal system.

The mode of application of formulations on the skin (e.g., occlusion) may affect the delivery of the incorporated molecules into and across the skin [2]. For this reason, we tested the application of FITC Weblike under various occlusion conditions and measured the skin penetration of the probe by CLSM. The obtained data indicated that non-immediate occlusive application of the Weblike composition resulted in the highest penetration profile. One possible explanation for this observation is that during the short period of opened application, the high ethanol concentration and the small phospholipid reverse micelles enhance the probe delivery into the skin. Concomitantly the film is generated and then is occluded, leading to better penetration into the deeper layers of the skin.

In the present work, we also introduced the Cannaweb carrier as a special type of Weblike film-forming carrier containing HSO for further enhancement of the delivery of lipophilic molecules to the deeper skin layers. Results of the CLSM examination of the in vitro skin delivery of FITC demonstrated an enhanced penetration by ~2-fold into the deep skin layers from Cannaweb compared to a Weblike system not containing this additive. Moreover, the in vitro CBD permeation and penetration were increased by 7- and 11-fold, respectively, as a result of application in Cannaweb.

HSO oil contains polyunsaturated fatty acids, tocopherols, tocotrienols, phytosterols, phospholipids and carotenes [35,36]. Noteworthy, we are the first group incorporating HSO in a dermal/transdermal delivery system for enhancing the delivery properties. The enhancing penetration mechanism into the skin of this oil should be further investigated.

We have further evaluated the antinociceptive effect of CBD applied from Cannaweb as compared to a Weblike system lacking oil in a pain mice model. We found that 1 h following treatment, the Cannaweb system containing 3% HSO led to efficient analgesia expressed by a high MPE value of 67.2%. Noteworthily, an equal dose of CBD applied in the Weblike not containing HSO, resulted in an MPE value of only 36.0%. This strong analgesic effect of CBD when applied in Cannweb points towards the suitability of this film forming system for pain treatment.

In a world-first publication on CBD transdermal delivery, Lodzki et al. [37] incorporated the molecule in ethosomes and evaluated its anti-inflammatory effect in a carrageenan induced sub-planter edema. The reported results indicated that the application of 100 mg ethosomal patches containing 3 mg CBD, for 19 h, prevented inflammation and edema.

Several publications in the literature investigated the film-forming systems for dermal and transdermal delivery of active molecules. Rande et al. [15] prepared and investigated film forming sprays of Ropivacaine for pain alleviation. The systems comprised different grades of Eudragit^®^ and either ethanol or isopropanol as volatile solvents. The antinociceptive effect of the drug applied from the film forming system was evaluated on rats in comparison with a conventional Lidocaine gel using the hot plate test. The results indicated that the analgesic effect of the film forming systems was slightly better than the conventional Lidocaine gel. However, the results were statistically not significant (*p* ˃ 0.05). Other works added penetration enhancers to the film-forming system to enhance the skin delivery of the incorporated active molecules and improve their pharmacological actions. Ammar et al. [12] used oleic and linoleic acids, Capryol^®^90 (PGMC), Lauroglycol^®^90 (PGML), Transcutol^®^ P (TNS), Azone^®^ and D-limonene in Ketorolac film-forming systems to improve the drug penetration through porcine skin. The highest permeation efficiencies of 59.6 and 31.6% resulted from adding Lauroglycol^®^90 and Azone^®^, respectively. These systems showed an improved analgesic effect of Ketorolac by up to 2-fold compared to the film-forming system containing oleic acid, as observed in the hot plate test in rats.

One promising application of Weblike systems is in nail delivery. In a preliminary experiment, we tested the ability of a Weblike formulation to facilitate molecules permeation through porcine nail using Franz diffusion cells and multiphoton microscopy. The results (not shown) indicated that the Weblike system delivered Rhodamine 6G into the nail to a depth of 350 µm.

The safety of carriers for dermal/ transdermal delivery systems is a crucial parameter. In our first publication [17], the local safety of the Weblike dermal/transdermal delivery carrier was assessed in an in vitro experiment using a reconstructed human epidermis, EpiDerm, a skin model closely parallels human skin. The results indicated that the new carrier had no effect on cell viability, suggesting its safety and the absence of any irritant effect of the carrier.

## 5. Conclusions

Weblike and Cannaweb film-forming formulations investigated in this work show an enhanced dermal and transdermal delivery of molecules into and across the skin. An improved analgesic effect of CBD in a pain mice model was achieved by means of topical application in Cannaweb. These data encourage clinical studies using these systems. This new approach may also open the way for better and noninvasive treatment of severe pain.

## Figures and Tables

**Figure 1 pharmaceutics-15-00397-f001:**
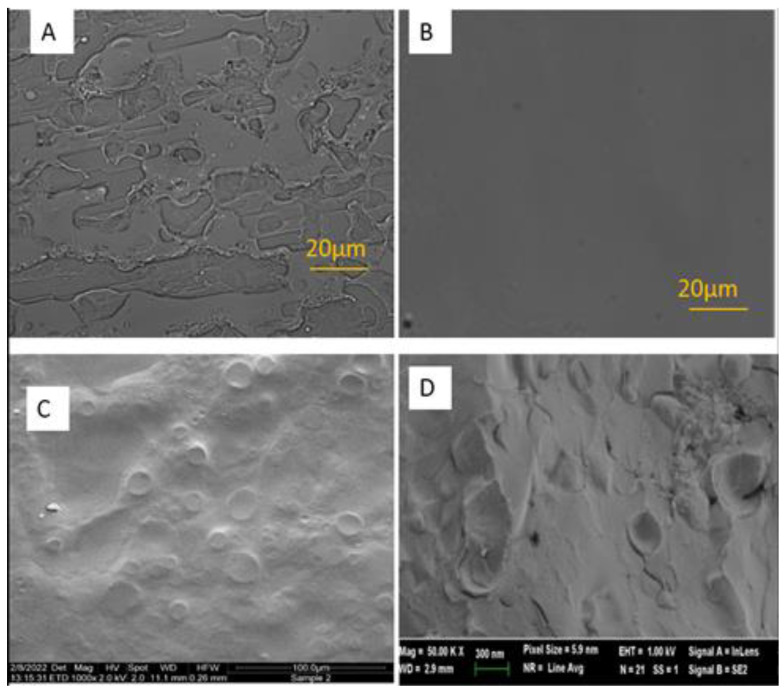
Structured Weblike film obtained from: (**A**) Weblike carrier (F9) by light microscopy using ×40 objective lens; (**B**) Control system not containing phospholipid by light microscopy), using ×40 objective lens; (**C**) Weblike carrier (F4) by SEM ×500; (**D**) Weblike carrier (F2) by CryoSEM. For the carriers’ compositions, see Table 1.

**Figure 2 pharmaceutics-15-00397-f002:**
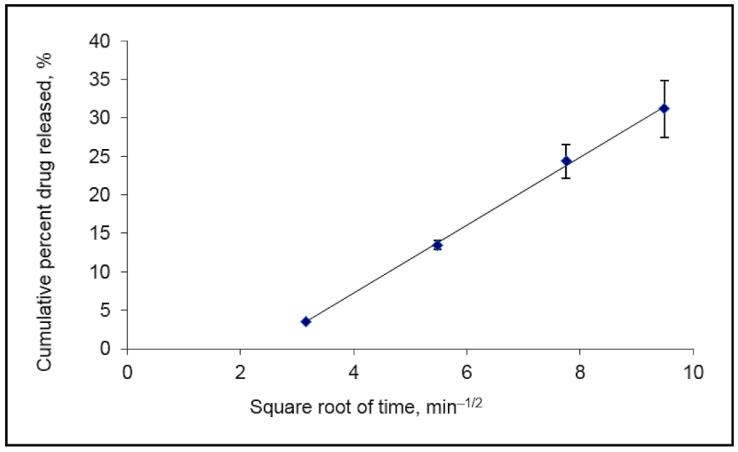
In vitro release profile of Lidocaine from Weblike film according to Higuchi’s plot. *n* = 5, Mean ± SD.

**Figure 3 pharmaceutics-15-00397-f003:**
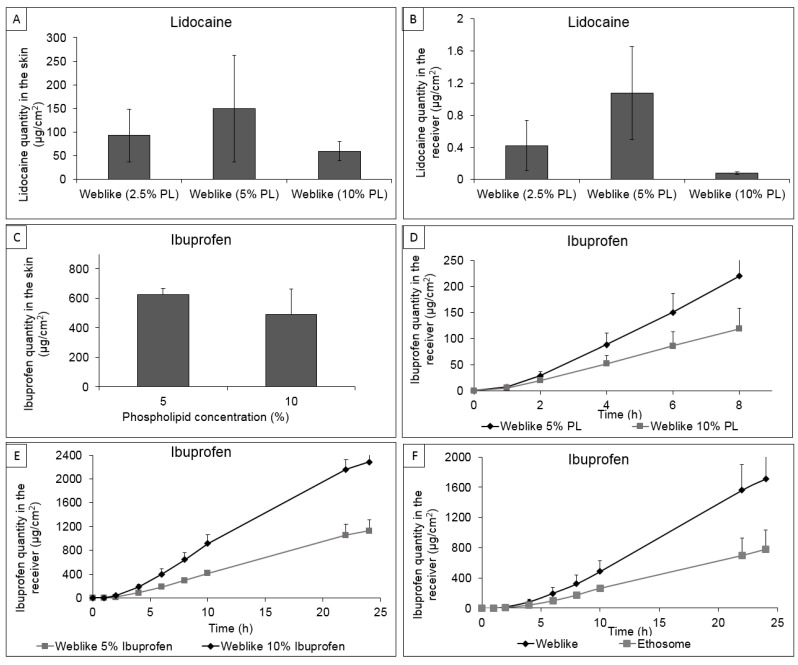
In vitro delivery into or across the skin of Lidocaine or Ibuprofen non-occlusive application of Weblike systems: (**A**) Lidocaine penetration into the skin 30 min following application in Weblike systems containing various phospholipid concentrations (F10–F12); (**B**) Lidocaine permeation across the skin 30 min following application in Weblike systems containing various phospholipid concentrations (F10–F12); (**C**) Ibuprofen penetration into the skin during 8 h following Weblike systems containing various concentrations of phospholipid (F14, F15); (**D**) Ibuprofen permeation across the skin during 8 h following application of Weblike systems containing various phospholipid concentrations (F14–F15); (**E**) Ibuprofen permeation across the skin during 24 h following application of Weblike systems containing various Ibuprofen concentrations (F13, F14) (**F**) Ibuprofen permeation across the skin during 24 h following application in Weblike system (F14) and in Ethosome. *n* = 5, Mean ± SD.

**Figure 4 pharmaceutics-15-00397-f004:**
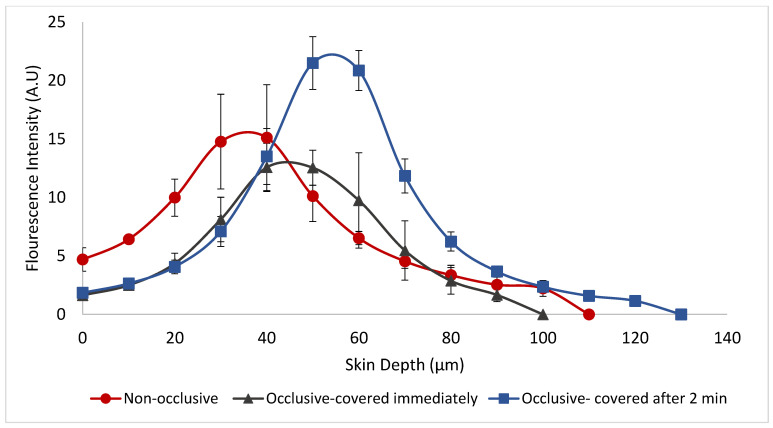
Mean fluorescence intensity (AU) of FITC measured following application of F7 (Cannaweb carrier) containing the probe applied: 1. Non-occlusively, 2. Occlusively, covered immediately, 3. Occlusively, covered 2 min after application, Mean ± SEM.

**Figure 5 pharmaceutics-15-00397-f005:**
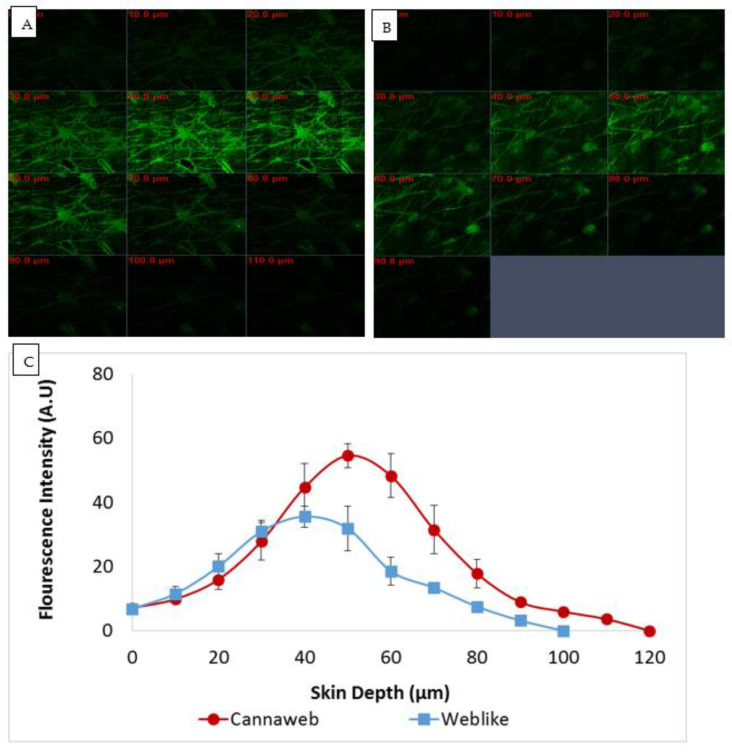
FITC delivery into the skin by CLSM: (**A**) Representative CLS micrographs of area skin sections at various depths treated with FITC in F7 (Cannaweb with HSO), (**B**) Representative CLS micrographs of area skin sections at various depths treated with FITC in F6 (Weblike without HSO), (**C**) Mean fluorescence intensity (AU) measured at various skin depths following application of Cannaweb with HSO vs. Weblike without HSO (*n* = 4), Mean ± SEM. *p* < 0.05 (considered significant) at 50–70 µm and 100 µm, *p* < 0.01 (considered very significant) at 80 and 90 µm, by One-Way ANOVA.

**Figure 6 pharmaceutics-15-00397-f006:**
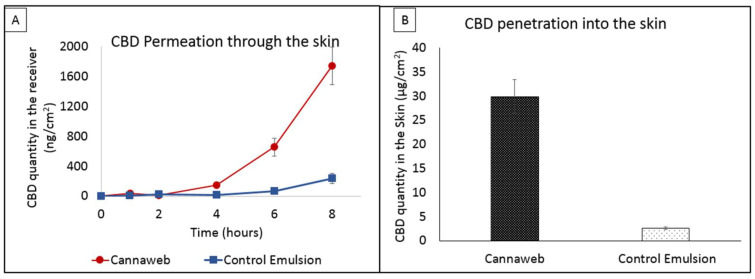
In vitro profiles of CBD (**A**) permeation across the skin (**B**) penetration into the skin following application from F16 (Cannaweb system) vs. Control emulsion containing an equal CBD amount. *n* = 4, Mean ± SD, *p* < 0.01 considered very significant by unpaired t-test.

**Figure 7 pharmaceutics-15-00397-f007:**
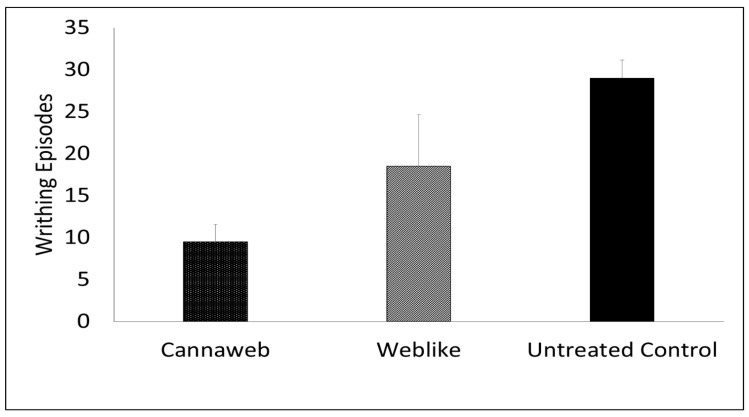
Mean writing counts in mice treated with 100 mg/kg CBD topically from F16 (Cannaweb containing 3% HSO), (*n* = 5) and F17 (Weblike not containing HSO), (*n* = 4), each containing 5% *w*/*w* CBD. The treatment was carried out 1 h prior to IP injection of acetic acid and, compared to untreated control mice, received an IP injection of acetic acid (*n* = 4) (Mean ±SD). *p* < 0.01 (considered very significant) for Cannaweb containing HSO vs. Untreated control group, *p* < 0.05 (considered significant) for Cannaweb containing 3% HSO vs. Weblike not containing HSO, by One-Way ANOVA.

**Table 1 pharmaceutics-15-00397-t001:** Composition of Weblike carrier formulations.

Weblike Carrier Code	Concentration *w/w*%							
	PL90G	ETOH	Ethyl Acetate	Klucel ^®^ HF	PG	HSO	Vit E	Water
F1	5	60	-	0.7	-	-	-	To 100
F2	5	65	-	0.7	-	-	-	To 100
F3	5	70	-	0.7	-	-	-	To 100
F4	7.14	71.43	-	2.14	7.14	-	-	-
F5	8.85	88.5	-	2.65	-	-	-	-
F6	2	97.5	-	0.5	-	-	-	-
F7	2	94.5	-	0.5	-	3	-	-
F8	1	98	-	0.5	-	-	0.5	-
F9	1	90.5	3	0.5	-	5	-	-

**Table 2 pharmaceutics-15-00397-t002:** Composition of Weblike systems containing drugs.

Weblike System Code	Concentration *w/w*%										
	PL90G	ETOH	Klucel ^®^ HF	HSO	Vit E	PG	Lidocaine Base	Ibuprofen	CBD	Finasteride	Water
F10	2.5	65	0.7	-	-	-	9.6	-	-	-	To 100
F11	5	65	0.7	-	-	-	9.6	-	-	-	To 100
F12	10	65	0.7	-	-	-	9.6	-	-	-	To 100
F13	5	65	0.7	-	-	-	-	5	-	-	To 100
F14	5	65	0.7	-	-	-	-	10	-	-	To 100
F15	10	65	0.7	-	-	-	-	10	-	-	To 100
F16	2	89.5	0.5	3	-	-	-	-	5	-	-
F17	2	92.5	0.5	-	-	-	-	-	5	-	-
F18	1	96	0.5	1	-	-	-	-	1	-	-
F19	1	97	0.5	-	0.5	-	-	-	1	-	-
F20	2.5	79.9	0.4	-	0.2	16	-	-	-	1	-

**Table 3 pharmaceutics-15-00397-t003:** Viscosity and pH of Weblike carrier at zero time, after 30, 60 and 90 days at RT, 4 °C and 40 °C, Mean ± SD.

	Zero Time	30 Days			60 Days			90 Days		
Temp. °C	RT	RT	4 °C	40 °C	RT	4 °C	40 °C	RT	4 °C	40 °C
Viscosity, cP	353 ± 17	387 ± 6	389 ± 4	315 ± 21	443 ± 22	440 ± 25	329 ± 4	333 ± 2	301 ± 18	316 ± 46
pH	4.77 ± 0.11	4.88 ± 0.06	4.82 ± 0.01	5.07 ± 0.05	4.96 ± 0.12	5.01 ± 0.13	5.12 ± 0.20	4.94 ± 0.03	4.85 ± 0.04	5.21 ± 0.21

## Data Availability

Not applicable.

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
