# Peer review of "Film Forming Systems for Delivery of Active Molecules into and across the Skin"

_pharmaceutics, 2023, doi:10.3390/pharmaceutics15020397_

Round 1

Reviewer 1 Report

The manuscript aimed to investigate and characterized a film-forming system previously published. Respecting the previous article, the authors aimed to discuss how components' concentration and skin application impact drug permeation. The topics might be of interest; however, due to several relevant weaknesses (especially in the experimental setup), the manuscript is not suitable for being published in Pharmaceutics.

  1. The authors proposed an alternative technological platform containing hemp seed oil (HSO) and compared its performance with HSO-free formulations. The HSO should be characterized in-depth. In particular, the concentration of CBD in HSO should be assessed. This aspect is particularly relevant to interpreting the in vivo results. 
  2. The drug-loaded formulations contained in Table 2 are not coherent with the placebo ones reported in Table 1. 
  3. Ethosomes are used in some IVPTs. No information on their preparation and characterization is reported in the manuscript.
  4. The IVPT protocols are poorly described. For example, sampling times, and skin integrity assessment are missing. No control formulations (e.g., drug solutions/suspension) were included. It seems that different experiments are performed independently and, then, merged together.
  5. The in vitro permeation profile of CBD should be carried out.
  6. The results of both retention and permeation studies should be expressed as mass normalized by permeation area.  
  7. In the discussion, the authors reported that films were in terms of DLS (lines 450-454). However, neither results have been reported previously, nor methods to obtain micelle after film dissolution were described.

Author Response

We thank the reviewer for the constructive comments. We have made amendments and additions as per his/her recommendations.

The manuscript aimed to investigate and characterized a film-forming system previously published. Respecting the previous article, the authors aimed to discuss how components' concentration and skin application impact drug permeation. The topics might be of interest; however, due to several relevant weaknesses (especially in the experimental setup), the manuscript is not suitable for being published in Pharmaceutics.

Point 1: The authors proposed an alternative technological platform containing hemp seed oil (HSO) and compared its performance with HSO-free formulations. The HSO should be characterized in-depth. In particular, the concentration of CBD in HSO should be assessed. This aspect is particularly relevant to interpreting the in vivo results. 

Response 1: CBD concentration in the HSO we used in these experiments is given: Page 16, lines 867-869 in the revised manuscript with track changes. It should be noted that a very negligible concentration of CBD is found (1.4µg/ml, 0.000142%) which could not affect any behavior or results we obtained.

Point 2: The drug-loaded formulations contained in Table 2 are not coherent with the placebo ones reported in Table 1.

Response 2: Table 1 contains placebo formulations with various component concentrations we tested to learn the behavior the carrier. In Table 2 we present the corresponding formulations containing active molecules. It should be noted that the difference in the ethanol concentration is a result of the addition of the active molecule keeping the total ingredient concentration to 100%.

Point 3: Ethosomes are used in some IVPTs. No information on their preparation and characterization is reported in the manuscript.

Response 3: Information about ethosomes is now given: Page 6, Lines 215-217, and Page 14, line 396 in the revised manuscript with track changes.

Point 4: The IVPT protocols are poorly described. For example, sampling times, and skin integrity assessment are missing. No control formulations (e.g., drug solutions/suspension) were included. It seems that different experiments are performed independently and, then, merged together.

Response 4: Requested details are now added to the protocols: Sections 2.3.4. and 2.3.5. Pages 5-6, in the revised manuscript with track changes.

An experiment with a control formulation is added: Sections 2.3.7, pages 6, 7 and 3.7. Page 17, in the revised manuscript with track changes.

Dear reviewer, this manuscript contains important data on a new film forming carrier for dermal and transdermal drug delivery. We believe that it will be of interest for many researchers working in the field and will encourage further research on this topic.

Point 5: The in vitro permeation profile of CBD should be carried out.

Response 5: The in vitro permeation profile of CBD is added: Sections 2.3.7, pages 6, 7 and 3.7. Page 17, in the revised manuscript with track changes.

Point 6: The results of both retention and permeation studies should be expressed as mass normalized by permeation area.  

Response 6: Corrected: Section 3.4., pages 11- 13 lines 617-619, 715-716 and 729-731 and in Figure 3, page 12,  in the revised manuscript with track changes.

Point 7: In the discussion, the authors reported that films were in terms of DLS (lines 450-454). However, neither results have been reported previously, nor methods to obtain micelle after film dissolution were described.

Response 7: These results are reported, Section 3.2., in the original manuscript.

Reviewer 2 Report

This report investigates polymeric carriers for administration of active molecules into and across the skin.

The experimentation was done in a systematic manner and the report was written fairly well. The manuscript can be further improved after major revisions suggested below:

In section 3.1 (results) it would be interesting to show how the results of the development and characterization experiments showed the formulations with the greatest potential. What were the parameters used in this step to determine the formulation with the greatest potential?

In section 3.3. I think it would be interesting to show the release of the 3 actives (lidocaine, ibuprofen and cannabidiol. Or just use a model drug for all the experiments in the manuscript.

Line 286-293. The film's release kinetics were unclear. Higuchi or first order?

Line 308: it was not clear whether the respectively refers to the concentration of 2.5 and 5.0% or 5 and 10%.

line 324, why didn't you use the same concentrations of phospholipids in the studies with ibuprofen?

Still in this section (3.4), why not evaluate the penetration of cannabidiol, since it will be studied in the next section?

Line 441-448. The discussion brings reports of the methodology. It would be more interesting to discuss the results and not just bring up the discussion of the results.

In short, the manuscript is a little messed up. Three molecules are investigated in different tests. It would be more interesting to use a model drug molecule and conduct the entire experimental part with that molecule, or to complement the studies for all model molecules studied, which would make the manuscript too long.

Reviewer 3 Report

Ladies and Gentlemen,

The work is very interesting and contains a lot of data and research methods. Nevertheless, it requires a few corrections.

Introduction

The introduction lacks a clear purpose of the work and its meaning, including specifically tested hypotheses (lines 63-77). Make sure that the introduction is understandable to scientists working outside the topic of the work.

Minor mistakes:

line 43, 45 - number cannot start a sentence

line 55 - we don't start a sentence with for example

Materials and Methods

The number of repetitions performed should be indicated in each method.

Results

Captions under drawings do not have to contain some technical data. They should be in Methods.

Figure 5 on the left is redundant letter C.

Round 2

Reviewer 1 Report

The manuscript has been significantly improved based on the reviewers’ comments. However, some additional majors should be addressed before considering it for publication. 

  1. The CBD assay of HSO should be clearly reported in the material section. 
  2. The ethosomes' composition, preparation methods and characterization should be reported in "methods" section. 
  3. The terminology used in all Figures and Tables should be harmonized since the current version may induce misleading interpretations. For example, Figures 6 and 7 referred to experiments carried out with F16 (containing 5% of CBD), but the Figures’ legends reported Cannaweb (3% HSO). Based on the text, the observed results cannot be related to 3% HSO, but to the loaded CBD. The authors are invited to use codes reported in Tables 1 and 2 to identify all formulations throughout the manuscript. 
  4. The mass balance should be provided for in vitro permeation studies. For example, estimating an applied IBU dose of about 2.5mg (assuming Formulation density = water), it seems a little strange that 2400 mg/cm2 were permeated through the membrane after 24 hours. 
  5. Since experiments with different time sampling are reported in Figure 3, The sampling times of sections A, B, and C should be clearly stated in the Figure caption.

Round 3

Reviewer 1 Report

The manuscript has been reviewed according to the previous comments and suggestions. It seems suitable for publication.